# Fabrication of Ni$_2$P Cocatalyzed CdS Nanorods with a Well-Defined Heterointerface for Enhanced Photocatalytic H$_2$ Evolution

Mengdie Cai [1,*,†], Siyu Cao [1,†], Zhenzhen Zhuo [1], Xue Wang [1], Kangzhong Shi [2], Qin Cheng [1], Zhaoming Xue [1], Xi Du [2,*], Cheng Shen [1], Xianchun Liu [1], Rui Wang [1], Lu Shi [1] and Song Sun [1,*]

[1]  School of Chemistry and Chemical Engineering, Anhui University, Hefei 230601, China; csyahu2022@163.com (S.C.); zhuozhenzhen2020@163.com (Z.Z.); wxahu2020@163.com (X.W.); chengqin0928@163.com (Q.C.); zmxue@ahu.edu.cn (Z.X.); shen_cheng124@163.com (C.S.); lxc21226119@mail.ustc.edu.cn (X.L.); wangrui20220311@163.com (R.W.); slahu2022@163.com (L.S.)
[2]  Eastman Shuangwei Fibers Company Limited, Hefei 230601, China; shikz@esfcl.com
[*]  Correspondence: caimengdie1987@163.com (M.C.); dux@esfcl.com (X.D.); suns@ustc.edu.cn (S.S.)
[†]  These authors contributed equally to this work.

**Abstract:** Developing non-noble metal photocatalysts for efficient photocatalytic hydrogen evolution is crucial for exploiting renewable energy. In this study, a photocatalyst of Ni$_2$P/CdS nanorods consisting of cadmium sulfide (CdS) nanorods (NRs) decorated with Ni$_2$P nanoparticles (NPs) was fabricated using an in-situ solvothermal method with red phosphor (P) as the P source. Ni$_2$P NPs were tightly anchored on the surface of CdS NRs to form a core-shell structure with a well-defined heterointerface, aiming to achieve a highly efficient photocatalytic H$_2$ generation. The as-synthesized 2%Ni$_2$P/CdS NRs photocatalyst exhibited the significantly improved photocatalytic H$_2$ evolution rate of 260.2 μmol·h$^{-1}$, more than 20 folds higher than that of bare CdS NRs. Moreover, the as-synthesized 2%Ni$_2$P/CdS NRs photocatalyst demonstrated an excellent stability, even better than that of Pt/CdS NRs. The photocatalytic performance enhancement was ascribed to the core-shell structure with the interfacial Schottky junction between Ni$_2$P NPs and CdS NRs and the accompanying fast and effective photogenerated charge carriers' separation and transfer. This work provides a new strategy for designing non-noble metal photocatalysts to replace the noble catalysts for photocatalytic water splitting.

**Keywords:** H$_2$ evolution; photocatalysis; CdS nanorods; core-shell structure; Ni$_2$P nanoparticles

## 1. Introduction

The harnessing and conversion of solar energy into clear and green energy through photocatalytic water splitting is an ideal and attractive strategy to convert solar power into hydrogen energy [1–3]. However, it is still a challenge to obtain the high efficiency photocatalysts for photocatalytic hydrogen generation. To date, numerous photocatalysts have been developed and well explored, including metal oxides [4,5], sulfides [6,7], and nitrides [8,9]. Among them, cadmium sulfide (CdS) nanostructures are efficient for hydrogen production due to their appropriate band gap (2.43 eV) and catalytic functions [10]. It is well known that different structures have different influences on the photocatalytic performance of CdS catalysts. In particular, one dimensional (1D) CdS nanorods (NRs) have noticeably attracted considerable attention due to their inherent advantages for photocatalysis, including a large surface area with a high length-to-diameter ratio, highly anisotropic electronic properties to allow the fast separation of photogenerated charge carriers, and abundant active sites [11]. Consequently, fabricating 1D CdS NRs would improve the photocatalytic hydrogen production activity. Nevertheless, suffering from the instability by photo-induced corrosion and severe recombination of photogenerated

electron-hole pairs may lead to their poor stability and low activity, thereby hindering the wide application of pristine 1D CdS NRs [11,12]. Therefore, to address these issues, much work has been carried out to suppress the high recombination rate of photoexcited charge carriers and to improve the stability of CdS NRs, including establishment of heterojunctions, introduction of cocatalysts, and design of Z-type systems [13–15]. Typically, loading cocatalysts is one of the most effective and convenient methods for improving their catalytic activity toward photocatalytic hydrogen production, which can not only accelerate the separation of electron-hole pairs but also can provide active sites to promote the hydrogen generation reactions [16,17]. Generally, noble metals (e.g., Pt [18], Pd [19], and Au [20]) with a great work functionality can be deposited onto CdS NRs to enhance the activity via trapping electrons on metals to facilitate the hydrogen evolution by the Schottky barrier at the metal-semiconductors' interfaces. Nevertheless, the high price and scarcity limit the large-scale application of noble metals. Therefore, the development of non-noble metal cocatalysts with high efficiency has significantly attracted researches' attention.

To date, various non-noble metal cocatalysts, such as transition metal sulfide [21], transition metal carbide [22], and transition metal phosphides (TMPs) [23,24] have been investigated to enhance the photocatalytic hydrogen generation activity of CdS. Among them, TMPs have been widely explored and demonstrated to be among the most promising alternatives to Pt-based co-catalysts due to their merits of favorable electrical conductivity, metallic characteristics, reversible binding, and dissociation of hydrogen [24–26]. Specifically, the nickel phosphide ($Ni_2P$) can be a good cocatalyst to extract electrons from semiconductor with a lower $H^+$ reduction overpotential for hydrogen evolution [27], as Ni and P sites exposed on the surface of $Ni_2P$ (001) exhibit an ensemble effect, whereby proton-acceptor and hydride-acceptor centers can both accelerate the hydrogen evolution [28]. Various photocatalysts modified with $Ni_2P$ cocatalyst have been reported, such as $Ni_2P$-CdS [27], $Ni_2P$-g-$C_3N_4$ [29], $Ni_2P$/$TiO_2$ [30], and $Ni_2P$-$ZnIn_2S_4$ [31], and the experimental results have shown that $Ni_2P$ as the cocatalyst is an appropriate substitute for the noble metal Pt. At present, the synthetic methods of $Ni_2P$ are mainly involved calcination, hydrothermal, and photodeposition [29–31]. However, there are some disadvantages in these methods, such as hazardous reagent, poor repeatability, long time, and high energy. Therefore, it is highly essential to exploit a simple and convenient method for $Ni_2P$-based photocatalysts.

In the present study, $Ni_2P$ was loaded on the surface of CdS NRs via a solvothermal method with safe and nontoxic red phosphor (P) and nickel chloride ($NiCl_2$) as phosphororus and nickel sources in the solvent of ethanolamine. Herein, using ethanolamine as alkali solvent could provide $OH^-$, which reacts with red P to produce $e^-$, accompanied by the reduction of $Ni^{2+}$ to obtain Ni, the generated Ni subsequently reacted with red P to generate $Ni_2P$ [32,33]. The synthesis process of $Ni_2P$ could be described by the following reactions [32,33]:

$$HOCH_3CH_2NH_2 + H_2O \rightarrow HOCH_3CH_2NH_3^+ + OH^- \tag{1}$$

$$P + 7OH^- \rightarrow HPO_4^{2-} + 3H_2O + 5e^- \tag{2}$$

$$Ni^{2+} + 2e^- \rightarrow Ni \tag{3}$$

$$2Ni + P \rightarrow Ni_2P \tag{4}$$

During the synthesis process of $Ni_2P$/CdS NRs, after adding CdS NRs to the mixture solution with ethanolamine, red P and $NiCl_2$, $Ni^{2+}$ ions slowly electrostatically absorbed onto the surface of the negatively charged CdS NRs, the following solvothermal treatment results in the in-situ growth of $Ni_2P$ on the CdS NRs, which results in the intimate contact beween CdS and $Ni_2P$ [34]. The as-synthesized $Ni_2P$/CdS NRs with a core-shell structure shorten diffusion path of photogenerated charge carriers, expedite the transportation and separation of electron-holes pairs, and expose more active sites to produce $H_2$. The $Ni_2P$/CdS NRs composites exhibited excellent photocatalytic performance compared with

those of pure CdS NRs. Moreover, the $Ni_2P$ cocatalyst shell could effectively inhibit the photocorrosion of the core catalyst CdS NRs, subsequently improved the photocatalytic stability of CdS NRs. The current study reveals a new fabrication strategy of non-noble metal and high-effective photocatalyst.

## 2. Results and Discussion

### 2.1. Characterization of Photocatalysts

XRD patterns were collected to analyze the crystal structures of the samples. Figure 1a shows the XRD patterns of the as-synthesized m%$Ni_2P$/CdS NRs samples. All samples exhibit the similar diffraction patterns, in which the diffraction peaks at 24.9°, 26.72°, 28.24°, 37.00°, 43.82°, 48.09°, and 52.22° were referred to (100), (002), (101), (102), (110), (103), and (112) planes of hexagonal CdS (JCPDS No. 80-0006) with high crystallinity, respectively. However, the diffraction peaks of phosphide were not found in the XRD spectra of all $Ni_2P$/CdS NRs composites, which could be due to their low loading (≤8.0 wt.%) and high dispersion capabilities. As shown in Figure S1 (in the Supplementary Materials), for pure $Ni_2P$, the characteristic diffraction peaks located at 40.72°, 44.60°, 47.39°, and 54.22°, belong to the (111), (201), (210), and (300) planes of hexagonal $Ni_2P$ phase (JCPDS No. 74-1385), respectively, indicating the developed method for $Ni_2P$ synthesis is effective, which provides a guarantee for the successful synthesis of $Ni_2P$/CdS NRs samples. The further characterization analysis about the existence of $Ni_2P$ in $Ni_2P$/CdS NRs samples is essential.

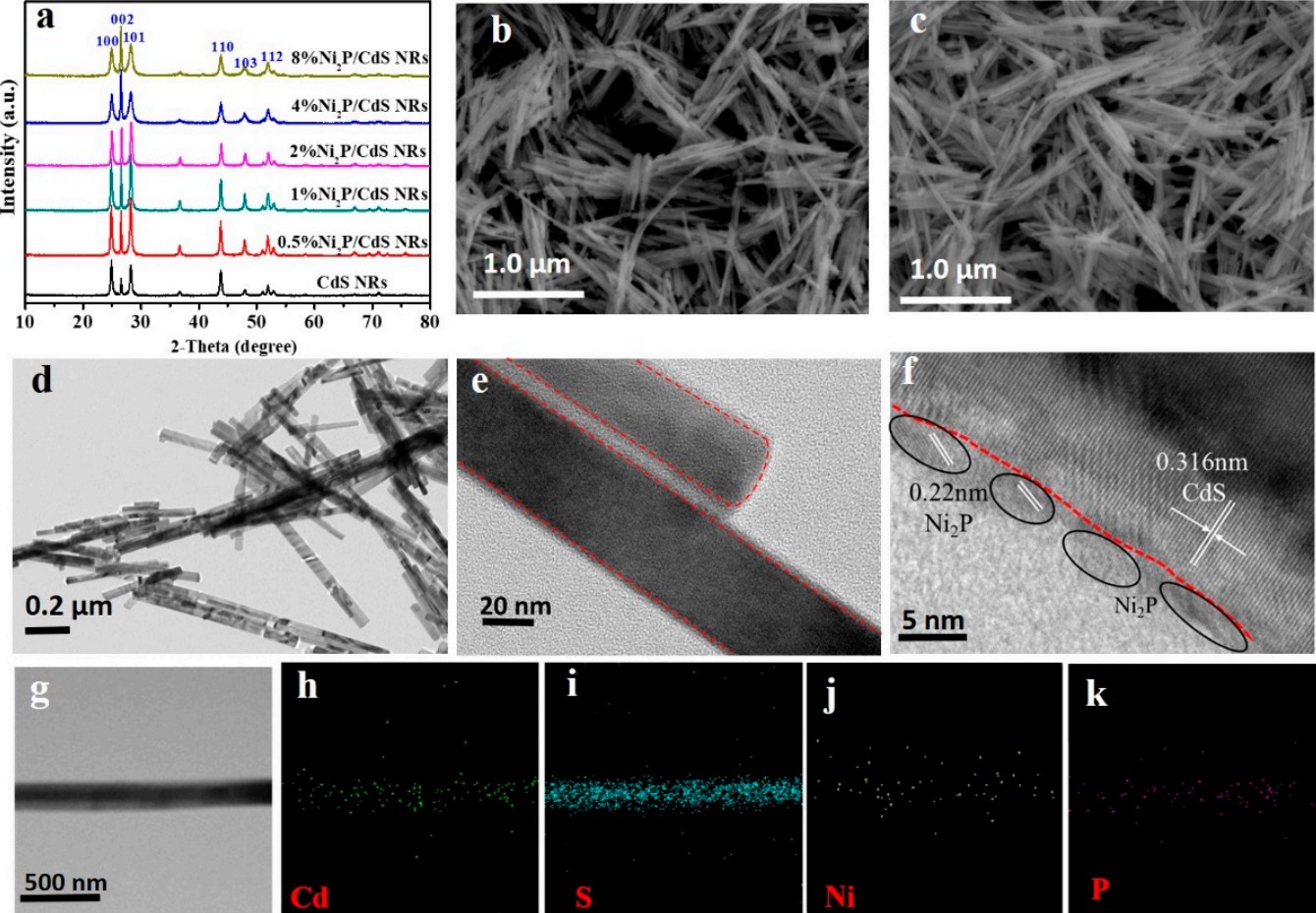

**Figure 1.** (**a**) XRD patterns of the CdS and m%$Ni_2P$/CdS NRs samples. (**b**) SEM image of CdS NRs; (**c**) SEM image; (**d**,**e**) TEM micrographs; (**f**) HRTEM image; (**g**–**k**) EDS mapping of the selected zone of 2%$Ni_2P$/CdS NRs sample in Figure 1g: (**h**) Cd; (**i**) S; (**j**) Ni; (**k**) P.

The morphologies of the CdS NRs, and 2%$Ni_2P$/CdS NRs heterostructure were investigated by the SEM. Figure 1b–d illustrate the structural features of the as-prepared samples. Pristine CdS NRs presented a clear 1D nanorods structure with the diameter range of 60–100 nm and the length range of 0.5–1.2 μm (Figure 1b). 2%$Ni_2P$/CdS NRs (Figure 1c) exhibited the similar morphology as that of pristine CdS NRs, indicating that after the phosphating treatment, the morphology structure of CdS NRs remains unchanged and $Ni_2P$ nanoparticles uniformly dispersed and anchored over CdS NRs. The TEM images (Figure 1e,f) reveal the core-shell structure of 2%$Ni_2P$/CdS NRs with tight interfacial contact between CdS NRs and $Ni_2P$ (red dashed line denoted the interface), which is beneficial for the well-defined heterostructure in 2%$Ni_2P$/CdS NRs. A close observation through high-resolution TEM (HRTEM) image presented in Figures 1f and S2 (in the Supplementary Materials), where the distinct lattice fringes of 0.316 and 0.220 nm could be corresponded to the CdS (002) and the $Ni_2P$ (211) planes, respectively, indicating the shell is $Ni_2P$ indeed. Furthermore, the elemental mapping of 2%$Ni_2P$/CdS NRs (Figure 1g–k) is used to confirm the existence of $Ni_2P$ and the core-shell structure of a single nanorod. As displayed in Figure 1h–k, Ni and P are uniformly distributed over the whole surface of the rod, indicating the close contact between $Ni_2P$ and CdS NRs, the result is consistent with the previous TEM and HRTEM analysis. The intimate contact interface between $Ni_2P$ and CdS NRs in the 2%$Ni_2P$/CdS NRs is beneficial for facilitating the photogenerated charge carriers' separation and transfer.

To probe the chemical composition and valence states of 2%$Ni_2P$/CdS NRs, X-ray photoelectron spectroscopy (XPS) was employed and the results showed in Figure S3 (in the Supplementary Materials) and Figure 2. The XPS survey spectrum of 2%$Ni_2P$/CdS NRs in Figure S3 (in the Supplementary Materials) revealed the presence of Cd, S, Ni, and P in this composite. The high-resolution XPS spectra of Cd 3d, S 2p, Ni 2p, and P 2p are illustrated in Figure 2. Figure 2a denoted the Cd 3d spectrum of CdS NRs and 2%$Ni_2P$/CdS NRs, the main peaks at 404.3 and 411.2 eV for CdS NRs, and at 404.7 and 411.7 eV for 2%$Ni_2P$/CdS NRs are corresponding to Cd $3d_{5/2}$ and Cd $3d_{3/2}$ of $Cd^{2+}$, respectively. In Figure 2b, the binding energies at 160.9 and 162.2 eV for CdS NRs, and at 161.4 and 162.7 eV for 2%$Ni_2P$/CdS NRs were ascribed to S $2p_{3/2}$ and S $2p_{1/2}$ of $S^{2-}$, respectively [35]. Compared with those of CdS NRs, the corresponding peaks of S 2p and Cd 3d in $Ni_2P$/CdS NRs moved toward higher binding energies (about 0.5 eV), indicating the strong electron interactions result from the close interface interaction between CdS NRs and $Ni_2P$. For the Ni $2p_{3/2}$ of 2%$Ni_2P$/CdS NRs, XPS peak can be deconvolved into two peaks centered at 855.2 and 857.3 eV (Figure 2c), which are assigned to the $Ni^{\delta+}$ and $Ni^{2+}$ in the $Ni_2P$, respectively [36]. Moreover, as observed in Figure 2d, in the P 2p spectrum of $Ni_2P$, the peak at 129.5 eV is ascribed to the $P^{\delta-}$ of Ni2P, while the peak at 133.8 eV to phosphate species due to the surface oxidation [32]. The above XPS analysis further reveals the existence of $Ni_2P$ and the intimate interface interaction between CdS NRs and $Ni_2P$ in 2%$Ni_2P$/CdS NRs composite.

## 2.2. Photocatalytic Activity Studies

The photocatalytic $H_2$ evolution activity of different $Ni_2P$/CdS NRs composites were tested under 300 W Xe lamp irradiation (λ > 420 nm) using 0.25 M $Na_2SO_3$ and 0.35 M $Na_2S$ as sacrificial agents (Figure 3a). As a control, Pt/CdS NRs and $Ni_2P$ were investigated for making comparison. The photocatalytic hydrogen evolution rate of pure CdS NRs was 13.8 μmol·$h^{-1}$. After loading $Ni_2P$ cocatalyst on the surface of CdS NRs, the photocatalytic activity was significantly increased. The optimal $Ni_2P$ loading was 2% with a volcano tendency. The 2%$Ni_2P$/CdS NRs demonstrated an excellent $H_2$ generation rate (260.2 μmol·$h^{-1}$), which is nearly 20 folds higher than that of pure CdS NRs, and even higher than that of Pt/CdS (98.5 μmol·$h^{-1}$). The enhancement mechanism can be explained by that the quickly separation and transfer of photogenerated electrons from CdS NRs to $Ni_2P$ based on the interfacial contact within the core-shell structure of 2%$Ni_2P$/CdS NRs, thereby boosting the enhanced photocatalytic activity for hydrogen production. When the

Ni$_2$P loading content increased to even higher than 2%, the rate of H$_2$ evolution decreased, probably since the excessive Ni$_2$P cause the shielding effect for light absorption, which may lead to the decreased photocatalytic activity. The superior photocatalytic hydrogen evolution performance of 2%Ni$_2$P/CdS NRs is comparable or better than other reported CdS based photocatalysts [37–41] (Table S1 in the Supplementary Materials). The effectiveness of the Ni$_2$P cocatalyst was also demonstrated on CdS nanoparticles (Figure S4 in the Supplementary Materials). These data suggested the potential application of Ni$_2$P as an alternative cocatalyst for the replacement of expensive and scarce Pt due to its high efficiency and cost-effectiveness. It should be noted that bare Ni$_2$P exhibited no photocatalytic H$_2$ activity, indicating that Ni$_2$P is an inert photocatalyst for hydrogen evolution. Furthermore, the photocatalytic experiment was performed with 2%Ni$_2$P/CdS NRs + M (the physical mixture of Ni$_2$P and CdS NRs, see the Supplementary Materials). As shown in Figure S5 (in the Supplementary Materials), compared with bare CdS NRs there is no obvious improvement of hydrogen evolution in 2%Ni$_2$P/CdS NRs + M, demonstrating that the intimate interface interaction between Ni$_2$P and CdS NRs in 2%Ni$_2$P/CdS NRs is essential to the photogenerated charge carrier separation and transfer and the photocatalytic performance, the fabrication of the heterojunction between Ni$_2$P and CdS NRs can facilitate the photogenerated charge carriers separation, thereby resulting in excellent photocatalytic H$_2$ production activity.

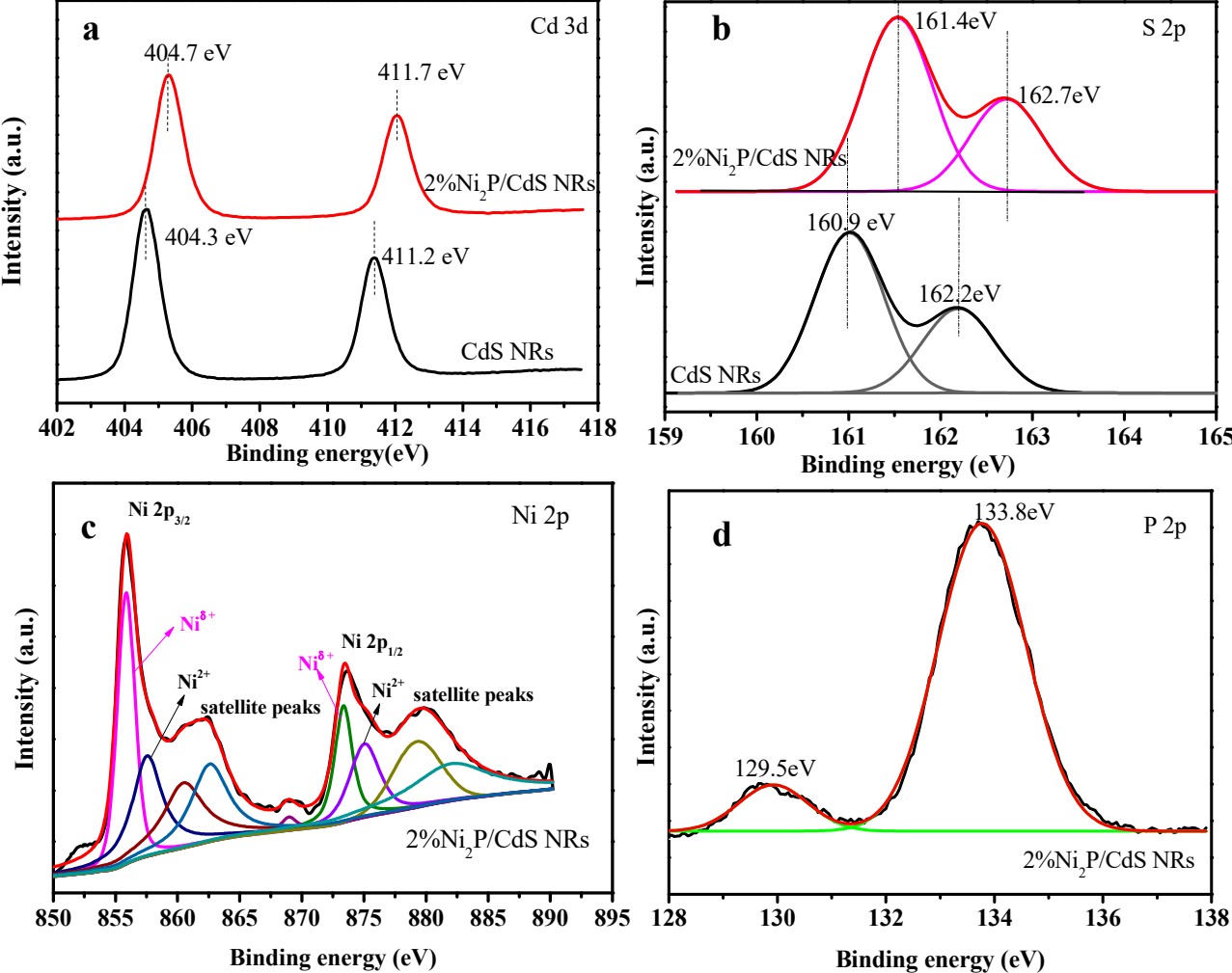

**Figure 2.** XPS spectra of (**a**) Cd 3d; and (**b**) S 2p of the CdS NRs and 2%Ni$_2$P/CdS NRs; (**c**) Ni 2p; and (**d**) P 2p of 2%Ni$_2$P/CdS NRs.

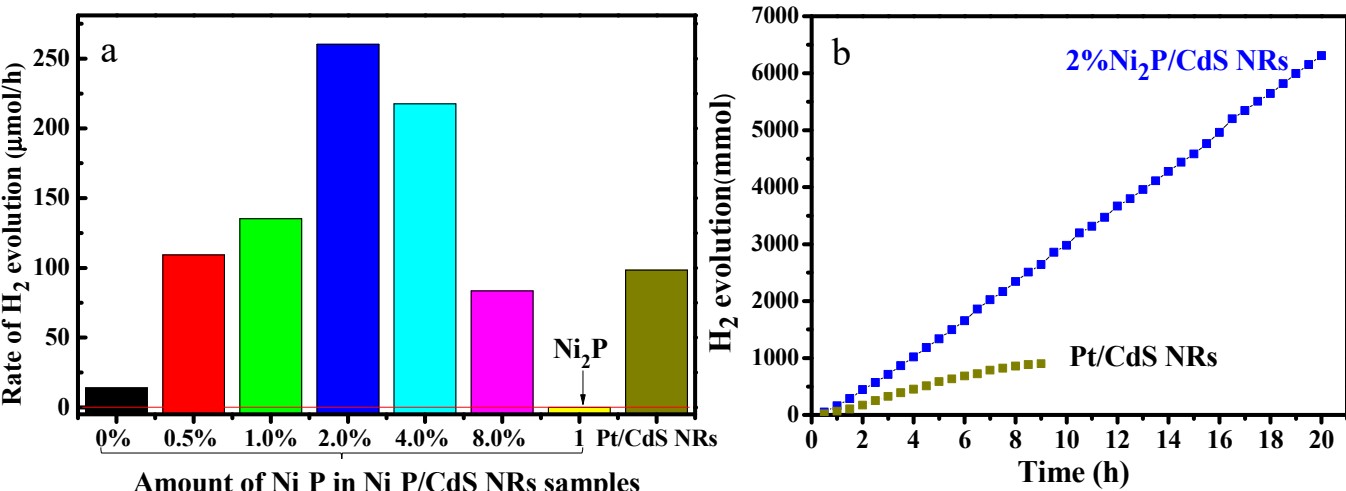

**Figure 3.** (**a**) The photocatalytic $H_2$ evolution rates of the m%$Ni_2P$/CdS NRs, $Ni_2P$, and Pt/CdS NRs samples under visible light irradiation ($\lambda \geq 420$ nm). (**b**) Long-term $H_2$ evolution for 20 h under visible light irradiation ($\lambda > 420$ nm) over 2 wt.% $Ni_2P$/CdS NRs and Pt/CdS NRs photocatalysts.

Photocatalyst stability is a decisive factor when a photocatalyst is practically used. The photocatalytic activity of 2%$Ni_2P$/CdS NRs and Pt/CdS NRs samples was investigated via long-time testing of photocatalytic $H_2$ evolution experiments, and the stability is compared in Figure 3b. The $H_2$ production rate did not display an obvious decrease even after being irradiated for 20 h, indicating that 2%$Ni_2P$/CdS NRs had a sufficient stability for photocatalytic $H_2$ production. Furthermore, the stability performance of 2%$Ni_2P$/CdS NRs was much better than that of Pt/CdS NRs, demonstrating the potential application of $Ni_2P$ as an alternative noble metal cocatalyst. This excellent photocatalytic stability could be ascribed to the following remarkable features: (1) the intimate contact between $Ni_2P$ and CdS NRs results in the efficient separation and transfer of photogenerated charges; (2) the $Ni_2P$ shell as the protective layer could effectively hinder the photocorrosion of CdS NRs [42].

### 2.3. Photocatalytic Mechanism

It is well known that the light-absorption ability, photogenerated charge carriers separation and transfer efficiency, and the surface reaction kinetics are the three most important factors affecting the photocatalytic performance. The light absorption properties of as-prepared $Ni_2P$/CdS NRs photocatalysts were investigated with UV-Vis DRS spectra (Figure 4a). The pristine CdS NRs exhibited visible light response with the absorption edge of about 520 nm, corresponding to an intrinsic bandgap value of CdS (2.4 eV). With the increasing content of $Ni_2P$, gradually increased light absorption intensity can be observed in the region of 500–800 nm, the absorption edges of the $Ni_2P$/CdS NRs samples were close to that of the pristine CdS, indicating that these Ni species or P species were not doped into the crystal structure of CdS NRs, in accordance with the XRD characterization.

To confirm the efficient photogenerated charge separation and transfer process in $Ni_2P$/CdS NRs composites, the photoluminescence (PL) emission spectrum and electrochemical measurements were employed. The PL spectra of CdS NRs and 2%$Ni_2P$/CdS NRs are presented in Figure 4b, it can be seen that the 2%$Ni_2P$/CdS NRs showed a decreased PL intensity compared with the CdS NRs. This result demonstrated that the deposition of $Ni_2P$ co-catalyst could efficiently suppress the photogenerated charge carriers' recombination of CdS NRs. Furthermore, the lifetime of photogenerated charge carriers is also an important factor in photocatalytic systems, which can be examined by the time resolved PL spectrum, the results are presented in Figure 4c. The calculated average lifetime (ave. $\tau$) for CdS was 6.75 ns, while that was 8.44 ns for $Ni_2P$/CdS NRs. The prolonged lifetime of charge carriers in 2%$Ni_2P$/CdS NRs can be ascribed to the promoted charge carriers' separation

and transfer efficiency, leading to the improved photocatalytic activity. Additionally, the transient photocurrent responses were measured to further investigate the migration and separation of photo-generated charge carriers (Figure 4d). It can be observed that the photocurrent response of 2%Ni$_2$P/CdS NRs exhibited a significantly higher intensity than that of pristine CdS NRs, indicating the efficient charge carrier separation and migration in 2%Ni$_2$P/CdS NRs. The above-mentioned results suggested that Ni$_2$P cocatalyst could effectively improve the photogenerated charge carriers' separation and transfer, causing the enhanced photocatalytic H$_2$ evolution performance.

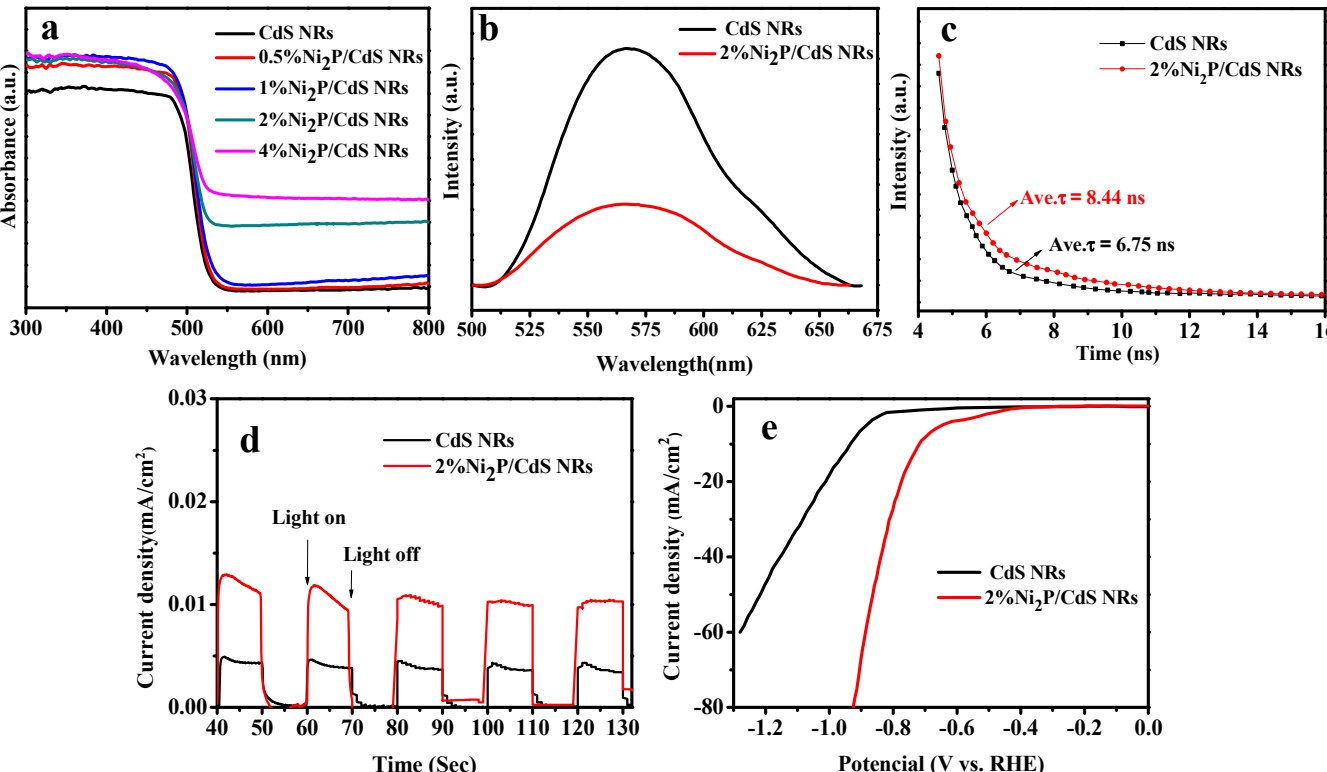

**Figure 4.** (**a**) UV-Vis diffuse reflectance spectra of CdS NRs and Ni$_2$P/CdS NRs composites with different Ni$_2$P loading amounts; (**b**) photoluminescence (PL) spectra; (**c**) time-resolved PL spectra (TRPL); (**d**) transient photocurrent responses; (**e**) polarization curves of the CdS NRs and 2%Ni$_2$P/CdS NRs samples.

LSV tests were herein employed to identify the H$_2$ evolution reaction (HER) kinetics of the pristine CdS and 2%Ni$_2$P/CdS NRs. As shown in Figure 4e, the 2%Ni$_2$P/CdS NRs has the lower overpotential compared with CdS NRs, indicating that Ni$_2$P cocatalyst can accelerate the photocatalytic HER kinetics, and subsequently boost the photocatalytic performance for H$_2$ production.

Based on the aforementioned experimental results, a probable photocatalytic mechanism for 2%Ni$_2$P/CdS NRs was proposed (Figure 5). Under visible light irradiation, electrons (e$^-$) are excited from the valence band (VB) of CdS NRs to its conduction band (CB), leaving holes (h$^+$) in the VB. As reported by Sun [43], the work function of metallic Ni$_2$P is lower than that of CdS NRs, at first the Ohmic contact between Ni$_2$P shell and CdS NRs core will be formed with the photogenerated e$^-$ transferring from Ni$_2$P to CdS NRs. When Ni$_2$P/CdS NRs is further exposed to irradiation, the accumulation of e$^-$ in the CB of CdS results in an upward-shift of CdS Fermi level to a new quasi-Fermi level [44], which drives a reverse transfer of e$^-$ from CdS to Ni$_2$P to participate the hydrogen evolution reaction, leading to a high photogenerated charge carrier separation and transfer efficiency [43,45]. Moreover, the photocorrosion of CdS NRs could be suppressed with the

Ni$_2$P shell loading, resulting in a robust photocatalytic hydrogen evolution rate and an excellent photostability.

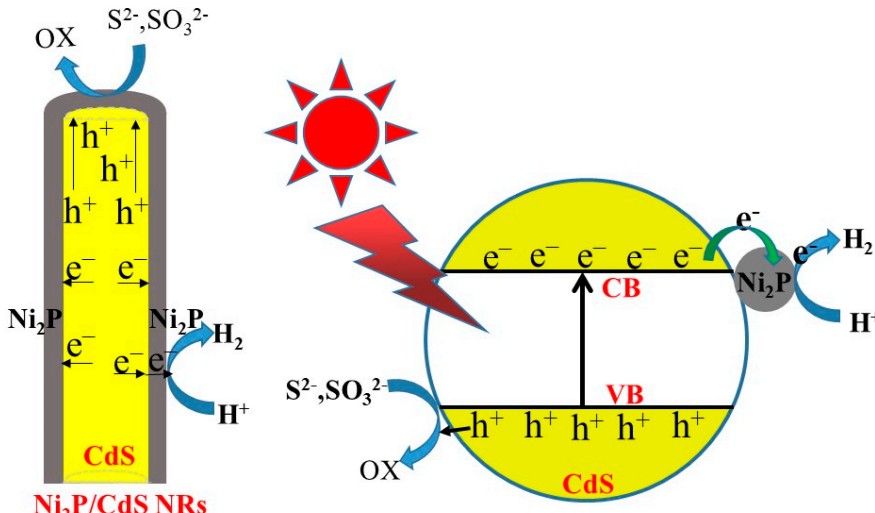

**Figure 5.** Proposed mechanism for photocatalytic H$_2$ production over 2%Ni$_2$P/CdS NRs under visible light irradiation. OX: oxidation; CB: conduction band; VB: valence band.

## 3. Materials and Methods

### 3.1. Sample Preparation

CdS NRs was prepared by the simple solvothermal method. Typically, 12.5 mmol of CdCl$_2$·4H$_2$O and 37.5 mmol of NH$_2$CSNH$_2$ were added into 60 mL of NH$_2$CH$_2$CH$_2$NH$_2$ with constant stirring for 2 h, then transferred to a 100 mL Teflon-lined autoclave and maintained at 160 °C for 24 h. Afterwards, the product was obtained by centrifugation and washed with distilled water and ethanol for 6 times and then dried at 60 °C for 24 h.

A Ni$_2$P/CdS NRs composite was prepared using the in-situ phosphorylation method. First, 200 mg of as-synthesized CdS NRs was added into a mixture of NiCl$_2$·6H$_2$O and red P (Ni:P molar ratio is fixed at 1:4) in 20 mL HOCH$_3$CH$_2$NH$_2$ with stirring for 30 min, then transferred to 100 mL Teflon-lined autoclaves and maintained at 180 °C for 10 h. The resultant Ni$_2$P/CdS NRs were washed with distilled water and ethanol for 6 times, and dried at 60 °C for 12 h. Here, the deposited content of Ni$_2$P in Ni$_2$P/CdS NRs composites can be adjusted by varying the primary amounts of red P and NiCl$_2$. The composites were labelled as m%Ni$_2$P/CdS NRs (m = 0, 0.5, 1, 2, 4, and 8), where m represents the mass ratio of Ni$_2$P to CdS NRs.

### 3.2. Characterization

X-ray diffraction (XRD) patterns were measured using a diffractometer (DX-2700 diffractometer; Bruker Co., Ltd., Berlin, Germany) with Cu K-$\alpha$ radiation ($\lambda$ = 0.15415 nm) operated at 40 kV and 4 mA, and recorded in a 2θ range of 10–80° with a step of 0.02°. The morphology and microstructure were observed by the scanning electron microscopy (SEM; Hitachi S-4800; Hitachi Co., Ltd., Tokyo, Japan) and transmission electron microscopy (TEM; JEM-2100; JEOL, Tokyo, Japan). The UV-Vis spectra were recorded via diffuse reflectance spectroscopy (DRS) of samples on a UV-3600 spectrophotometer (Hitachi Co., Ltd., Tokyo, Japan), in which BaSO$_4$ was used as the background. The photoluminescence (PL) spectra of the synthesized samples were taken using a F-4600 spectrometer (Hitachi Co., Ltd., Tokyo, Japan) at room temperature. In the case of X-ray photoelectron spectroscopy (XPS) (ESCALAB 250, Thermo Fischer Scientific, Waltham, MA, USA), an Al anode with a monochromator was used to significantly reduce the background signal. The binding energy was referenced to the C 1s peak taken at 284.8 eV.

*3.3. Photocatalytic Measurement*

Photocatalytic performance was evaluated in Labsolar-6A system (Perfect Light Co., Beijing, China) with a round upside-window for light irradiation. The light source adopted a 300 W Xe lamp ($\lambda \geq 420$ nm). Typically, 100 mg of the fabricated samples was dispersed into 100 mL solution containing $Na_2S$ and $Na_2SO_3$ as sacrificial reagents. The stirring was applied during the reaction to keep the samples in suspension status. The amount of hydrogen poduction was determined by gas chromatography (GC1690, Anhui Chromatography Co., Ltd., Hefei, China) equipped with a 5 Å molecular sieve column and a thermalconductivity detector (TCD).

## 4. Conclusions

In summary, $Ni_2P$ nanoparticles were successfully loaded on the surface of CdS NRs using a simple in-situ solvothermal method with red P as P source. With the deposition of an optimal amount of $Ni_2P$ nanoparticles, 2%$Ni_2P$/CdS NRs exhibited an outstanding efficiency for $H_2$ production (~260.2 $\mu mol \cdot h^{-1}$), which was significantly higher than that of 1 wt.% Pt/CdS NRs (98.5 $\mu mol \cdot h^{-1}$). In addition, 2%$Ni_2P$/CdS NRs also exhibited a promising stability. The intimate contact between $Ni_2P$ and CdS NRs could accelerate the photogenerated charge carriers' separation and transfer, and endow photocatalysts with excellent photocatalytic $H_2$ evolution performance.

**Supplementary Materials:** The following supporting information can be downloaded at: https://www.mdpi.com/article/10.3390/catal12040417/s1. Figure S1: XRD pattern of the $Ni_2P$; Figure S2: The HRTEM images of 2%$Ni_2P$/CdS NRs; Figure S3: The XPS survey spectra of 2%$Ni_2P$/CdS NRs; Figure S4: The photocatalytic $H_2$ evolution activity of as-prepared m%$Ni_2P$/CdS-H samples; Figure S5: The photocatalytic $H_2$ evolution activity of CdS NRs, 2%$Ni_2P$/CdS NRs + M and 2%$Ni_2P$/CdS NRs. Table S1: Comparison on the $H_2$ evolution performance of various CdS based photocatalysts. References [37–41] have been cited in the Supplementary Materials.

**Author Contributions:** Conceptualization, M.C. and S.S.; methodology, M.C., S.C., Z.Z., K.S., Q.C., Z.X., X.D. and X.L.; software, Z.Z. and Z.X.; validation, Q.C. and X.D.; data analysis, Z.Z., X.W., Z.X., C.S., R.W. and L.S.; literature research, S.C., X.W., K.S., C.S., X.L., R.W. and L.S.; resources, Q.C. and X.D.; data curation, M.C., S.C., X.W., C.S. and X.L.; writing—original draft preparation, S.C.; writing—review and editing, M.C. and S.S.; supervision, K.S. and S.S. All authors have read and agreed to the published version of the manuscript.

**Funding:** This research was funded by National Natural Science Foundation of China (Grant No. U1832165, 21902001 and 22102001), Anhui Provincial Natural Science Foundation (Grant No. 2008085QB85 and 2108085QB48), Key Research and Development Program of Anhui Province (Grant No. 202004a05020015 and 006233172019), and Higher Education Natural Science Foundation of Anhui Province (KJ2021A0027 and KJ2021A0029).

**Data Availability Statement:** The data presented in this study are available on request from the corresponding authors.

**Acknowledgments:** The authors acknowledge the financial support from Anhui University.

**Conflicts of Interest:** The authors declare no conflict of interest.

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
