# Peer review of "Fabrication of Ni2P Cocatalyzed CdS Nanorods with a Well-Defined Heterointerface for Enhanced Photocatalytic H2 Evolution"

_catalysts, doi:10.3390/catal12040417_

Round 1

Reviewer 1 Report

The proposed manuscript present a small but still acceptable amount of novelty for the use of red P as phosphorus source for Ni2P deposition on CdS has already been reported (RSC Adv 2021, 11, 12153) and the deposition of Ni2P on CdS nanorods is also described in Appl Catal B environ. 2021, 281, 119443. Then the combination of the two aspects (red P and CdS rods) is the novelty of the study.

On the scientific quality of the study itself, there are several aspects in the manuscript that require significant improvements.

- The last part of the introduction is awkward for balanced equations are given there without article reference and without any proof. Then the reference 32 quoted in the end of the introduction has nothing to do with the sentence it is related to.

- The scientific results hardly confirm the presence of Ni2P: Only one small peak in the 8% Ni2P/CdS diffractogram may tell about the presence of the crystalline phase. The TEM image quality is not good enough to check the presence of Ni2P (higher quality TEM images should be given in SI). The most problematic is the XPS analysis for the authors refer to an articles where the proposed value for Ni 2p and P 2p ar e different to that measured in the article. However the authors say that it is in agreement … Finally on the experimental point, the UV-Vis diffuse reflectance spectra are plotted with a strange vertical scale: intensity (a.u.) while it should be absorbance or reflectance. Moreover the shape of the curve is quite different from that of the article in Applied Catal B mentioned above.

- About the understanding of the system, authors first talk about a Z-scheme for the heterojunction and then in figure 5 they make a quite strange band alignment with the drawing of a nanoparticle of Ni2P rather than a band scheme for that semi-conductor (bandgap of 1.0 V). The band bending described is not in agreement with an electron transfer and again a careful read of the Applied Catalysis B article should help to understand how such a junction work.

Reviewer 2 Report

The manuscript describes synthesis of Ni2P cocatalyzed CdS for the photocatalytic hydrogen evolution under visible light. Although the topic itself is not new, this work is an example of a well-structured study with well-designed blank experiments and state-of-the-art sample characterization. I have only a few minor remarks.

  1. It is better to recalculate activity per hour and per gram of the photocatalyst.
  2. XRD patterns - all Miller indices should be labeled.
  3. XPS spectra - Ni2p peaks should be deconvoluted.
  4. The activity should be compared with recently published data on hydrogen evolution over CdS-based photocatalyst [e.g. 10.1016/S1872-2067(20)63597-5; 10.1002/chem.202002192; 10.1016/j.apcatb.2020.119853; 10.1016/j.ijhydene.2020.08.133; 10.1016/j.ijhydene.2021.03.227].

Round 2

Reviewer 1 Report

The authors still don't quote the very publication that make their work less original (Appl Catal B environ. 2021, 281, 119443).

The XPS and TEM analyses have been improved at an appropriate level.

The last point is the band scheme used in figure 5. The authors declare that Ni2P is a metal and refer to an article [38] that is not about Ni2P. That is the second time the reference used is inapropriate.

There are still ways to improve the manuscript 

Author Response

This manuscript is a resubmission of an earlier submission. The following is a list of the peer review reports and author responses from that submission.